# Identifying research priorities for pregnant South Asian immigrants in Canada: A James Lind Alliance approach

Anam Shahil-Feroz[1]*, Bibi Hajira[1], Sugandha Chandak[1], Salima Meherali[2], Saraswathi Vedam[3], Anushka Ataullahjan[4], Rohan D'Souza[5,6]

1 Arthur Labatt Family School of Nursing, Faculty of Health Sciences, The University of Western Ontario, London, Ontario, Canada, 2 Faculty of Nursing, University of Alberta, Edmonton, Alberta, Canada, 3 Birthplace Lab, Division of Midwifery, Faculty of Medicine, University of British Columbia, Vancouver, British Columbia, Canada, 4 School of Health Studies, Faculty of Health Sciences, The University of Western Ontario, Lindon, Ontario, Canada, 5 Department of Obstetrics and Gynaecology, McMaster University, Hamilton, Ontario, Canada, 6 Department of Health Research Methods, Evidence, and Impact, McMaster University, Hamilton, Ontario, Canada

* ashahilf@uwo.ca

## Abstract

### Background

Pregnant South Asian immigrants (PSAI) make up a significant portion of immigrants to Canada and face a higher risk of adverse pregnancy outcomes compared to other ethnic groups. This disparity is largely due to cultural and linguistic barriers to maternity care, including language difficulties, limited cultural sensitivity in healthcare services, and a lack of awareness about culturally tailored educational resources. Despite the growing number of PSAI in Canada, there is limited understanding of how to best support their healthcare and well-being. To address this gap, we aim to conduct a priority-setting exercise to identify key research priorities and establish a patient-oriented research agenda that will drive long-term, impactful research and ultimately improve maternal health outcomes for PSAI in Canada.

### Methods

This project follows the James Lind Alliance (JLA) priority-setting partnership (PSP) methodology, which includes forming a steering committee, identifying and verifying uncertainties, conducting an interim priority-setting phase, and holding a final work-shop. Participants will include first-generation South Asian immigrant women from Bangladesh, India, Pakistan, and Sri Lanka, as well as clinicians, researchers, and community/professional organizations from Ontario, Alberta, and British Columbia. Data will be collected through Zoom-based recorded working group sessions and an online ranking survey. Qualitative data will be analyzed using an inductive content analysis approach supported by NVivo software. Subgroup diversity (e.g., ethnicity,

**Data availability statement:** Please note that this is a study protocol and thus no datasets were generated or reported during the current study.

**Funding:** This research is funded by the Canadian Institutes of Health Research Planning and Dissemination Grant (Application No. 197315). The funders had no role in study design, data collection and analysis, decision to publish, or preparation of the manuscript.

**Competing interests:** The authors have declared that no competing interests exist.

gender, age, and geography) will be tracked across participant groups. Consensus on top research priorities will be achieved through a structured ranking process followed by a facilitated virtual workshop. The study began in May 2025 and is expected to conclude by January 2026, a timeline consistent with similar JLA PSP initiatives.

## Dissemination

All findings will be shared through a peer-reviewed publication and conference presentations for the scientific community, a lay summary for community organizations, and a video and infographic for patient participants. Community and professional organizations will also support the dissemination of findings through their networks and social media channels.

## Introduction

Pregnancy is a significant event not only in the lives of the pregnant individuals and their unborn child but also in those of their spouses, family, and community [1–3]. High-quality pregnancy care is essential for positive pregnancy outcomes and experiences [4]. Globally, the delivery of prenatal care remains inadequate [4–7], especially among immigrant individuals, influenced by cultural differences, language barriers, negative care experiences, and a lack of understanding of their specific healthcare needs [8–13]. In particular, immigrant individuals of South Asian ethnicity living in Western societies are at higher risk for adverse pregnancy outcomes compared with other ethnicities, including stillbirth, preterm birth, and gestational diabetes mellitus [14]. Studies show over 20% of immigrants receive inadequate prenatal care [15], and South Asian women in Ontario are at increased risk of serious preeclampsia (aOR: 1.30), particularly first-time mothers (aOR: 1.38) [16]. Although research increasingly focuses on immigrant populations in general, the distinct perinatal care needs and research priorities of pregnant South Asian immigrants (PSAI) remain largely under-investigated. For this study, PSAI specifically refers to first-generation South Asian immigrants from Bangladesh, India, Pakistan, and Sri Lanka who have been pregnant or are currently pregnant, and who experienced pregnancy after immigrating to Canada. While the South Asian region also includes countries such as Nepal, Bhutan, and the Maldives, our focus was limited to the four countries with the largest South Asian immigrant populations in Canada.

Canada, one of the world's most ethnically diverse countries, is home to over seven million foreign-born individuals [17]. It has the second-highest percentage of immigrants globally, with nearly one in four people being an immigrant (23.0%) [18]. Projections indicate that immigrants will comprise between 29.1% to 34.0% of Canada's total population by 2041 [17]. Immigrant groups are growing in Canada, with South Asians being the largest visible minority group and one of the fastest-growing immigrant populations, representing 7.1% of the total population [19,20]. Predominantly, South Asian immigrants settle in Ontario (ON), British Columbia (BC), and Alberta (AB) [19], making these provinces primary hubs for cultural diversity and

immigrant settlement. Systematic collection of sociodemographic data in Canada remains inadequate, making it challenging to determine the exact number of PSAI and accurately assess their health outcomes. However, A small but growing body of evidence suggests that PSAI make up a significant proportion of immigrants in Canada and are at greater risk for adverse pregnancy outcomes, such as gestational diabetes and preeclampsia than other immigrant groups, due to culturally specific factors such as restrictive gender norms, stigma around pregnancy complications, limited English proficiency, and underutilization of services that are not culturally or linguistically tailored [14–16,21].

Despite the large and growing number of PSAI in Canada, there remains limited understanding of their unique maternity care needs and a lack of research on how best to support them. Preliminary findings from our ongoing scoping review reveal intersecting barriers that hinder PSAI's access to and navigation of maternity care services, including structural, cultural, communication and informational, and health system trust and responsiveness barriers. Structural barriers include financial constraints [22], limited availability and access to social and health services [23], and logistical challenges such as transportation difficulties [24,25], all of which contribute to delays in care-seeking or disengagement from the healthcare system. Cultural barriers include acculturative stress among PSAIs [26], social isolation stemming from limited access to family or community support networks [27], the absence of culturally tailored perinatal education [23,28] and a lack of responsiveness to cultural preferences in maternity care settings, such as the desire for female care providers [28] and culturally appropriate care practices. Communication and informational barriers are characterized by limited health literacy [29,30], significant language barriers reported by 33–43% of South Asian women compared to 0% of Canadian-born women [23,27], and the scarcity of translated or culturally relevant educational materials [23,28]. These challenges impede understanding of care options, reduce confidence in decision-making, and hinder meaningful participation in care. Health system trust and responsiveness remain significant concerns. Although PSAI are at disproportionately higher risk for gestational diabetes [31], culturally adapted interventions for prevention and management are often underutilized, largely due to limited awareness and low trust in the healthcare system [24]. These barriers contribute to frustration and miscommunication between healthcare providers and patients, as well as significantly impact the mental health of immigrant pregnant individuals, particularly increasing risks of perinatal depression among subgroups such as Punjabi-speaking women with low acculturation or limited social support [26,32,33].

These findings highlight the need to move beyond a one-size-fits-all approach o immigrant maternal health. Researching immigrants as a homogeneous population is counterproductive, as they have differential risk profiles, diverse understandings of health and illness, and significant cultural and linguistic variation. As a result of limited culturally relevant support, PSAI often delay or avoid seeking care and often rely on informal sources such as social media (e.g., Soul Sisters Canada) and friends and family [9,23,29,34] for guidance on choosing healthcare providers (midwife vs. obstetrician), navigating conflicting medical advice, seeking emotional and psychological support, and discussing cultural preferences with healthcare providers.

Currently, there is no care model that supports culturally and linguistically sensitive maternity care for PSAI in Canada beyond the standard pregnancy care. Therefore, it is critical to establish a research agenda that outlines research priorities for supporting the health and well-being of PSAI in Canada. This project aims to identify the top research priorities for PSAI in Canada by gathering insights from patient partners, relevant community and professional organizations, and care providers, including obstetricians, and researchers. This study will seek to answer the following questions: What are the top research priorities for PSAI to improve health and well-being outcomes for this group? What issues in Canada's maternity care landscape matter most to PSAI and care providers? The study findings will be important for guiding future research directions and optimizing the allocation of resources to improve outcomes for PSAI and their family caregivers.

## Materials and methods

### Design

This project will be guided by the James Lind Alliance (JLA) approach [35], which includes the following steps: *(1)* forming a priority setting partnership (PSP) steering committee, *(2)* gathering potential research uncertainties, *(3)* summarizing

and verifying the research uncertainties, *(4)* completing an interim priority setting phase, and *(5)* holding a final priority setting workshop [35]. As a partnership-driven approach [36], the JLA is well-suited to identify relevant research priorities for PSAI through collaboration with patients, family caregivers, and care providers, researchers, community and professional organizations. The inclusive JLA approach has been successfully used to identify research priorities for several conditions, including diabetes [37], dementia [38], Parkinson [39], chronic kidney disease [40], inflammatory bowel disease [41], cystic fibrosis [42], and eating disorders [43]. Our study will use this approach to identify research priorities for this specific minority group – PSAI.

## Study timeline

The study will be conducted over 1.5 years. Below is a summary of the key phases, activities, and responsible parties:

| Phase | Timeframe | Key Activities |
|---|---|---|
| Planning & Preparation | May 2025 | Ethics approval, recruitment materials, training |
| Recruitment | June 2025 | Participant outreach and enrollment |
| Data Collection | June 2025 – April 2026 | Working group sessions and priority-setting activities |
| Data Analysis | May – August 2026 | Inductive content analysis and synthesis of findings |
| Knowledge Translation | Sept – Dec 2026 | Peer-reviewed publications; policy briefs for decision-makers; knowledge exchange forums and events with community organizations; multimedia resources (e.g., videos, voice messages); stakeholder round tables with funders and policymakers |

## Research setting and participants

Eligible participants will include first-generation South Asian immigrants from Bangladesh, India, Pakistan, or Sri Lanka, aged 18 years or older, who are currently pregnant or have experienced pregnancy in Canada within the past five years. Participants must have experienced pregnancy after immigrating and be able to communicate in basic English. Immigration status may include permanent or temporary residents, including international students. To ensure inclusivity, the study is open to all individuals assigned female at birth who identify as women or gender-diverse. PSAIs are able to opt out of the study without any consequences. In addition, obstetricians and/or maternal-fetal medicine specialists, researchers, community organizations, and professional organizations in Ontario, Alberta, and British Columbia will take part in the priority-setting exercise. Although formal sample size calculations are not typically conducted for JLA-PSP projects [35] our estimated participant numbers are informed by prior studies that applied the JLA approach with pregnant populations [44–48] and individuals with chronic health condition [40–42], as well as by available funding. We anticipate recruiting approximately 15 patient partners, 6–8 clinicians, 6–8 researchers, and 8–10 community and professional organizations to ensure sufficient diversity and meaningful engagement across stakeholder groups. We will also employ purposive sampling to capture a broad range of perspectives across key characteristics such as immigration status, education level, employment status, age, parity, and length of time since arrival. Several recruitment strategies will be employed. PSAI participants will be recruited through community organizations and informed about the study via social media platforms (e.g., flyers on LinkedIn) and healthcare providers. Clinicians, researchers, and community or professional organizations have already been identified through existing relationships and team members.

## Step 1. Forming a PSP steering committee (SC)

A PSP Steering Committee was established to oversee all project activities and ensure adherence to the JLA process. The committee was initially formed in July 2024 during the grant conceptualization phase and includes representation

from all key stakeholder groups: two patient partners, one community organization, two professional organizations, one clinician, and four researchers. Committee members contributed to the development of the grant proposal and endorsed the importance of identifying research priorities for PSAI in Canada. The committee will remain engaged throughout the project, meeting regularly, approximately every two months via Zoom for one hour or through interim email communications to stay updated on research progress. Meeting minutes will be recorded to document decisions, identify any required corrective actions, ensure alignment with the JLA methodology, and track the implementation of recommendations in future sessions. During the most recent meeting held on April 22, 2025, the committee decided to organize separate working group sessions for clinicians/researchers, patient partners, community organizations, and professional associations.

## Step 2: Gathering potential research uncertainties through working group sessions

Potential research priorities (termed "uncertainties" by the JLA [35]) for PSAI will be determined through separate working group sessions with 1) patient partners, 2) clinicians/researchers, and 3) community and professional organizations. At least three working group sessions will be scheduled with each of these groups (with approximately 7–10 participants per group). Information about these sessions will be disseminated through the steering committee, community organizations, care providers, and social media platforms (e.g., LinkedIn). These sessions will be held online via Zoom and guided by separate semi-structured discussion guides tailored for each group (S1-S3 Files). The guides will be informed by our ongoing scoping review on perinatal care for PSAI in Canada and will include open-ended questions. They will facilitate discussion on identifying research priorities in the following areas: barriers to accessing culturally and linguistically appropriate maternity care; mental health and emotional well-being during pregnancy and postpartum; social support during pregnancy; tailored support for PSAI for high-risk pregnancy conditions; perceived gaps in the current maternity care landscape; and opportunities to provide tailored maternity care for PSAI. The specific questions will vary by group. For example, patient partners will be asked about the challenges they face in receiving culturally sensitive care, whereas care providers will be asked about the challenges in providing such care and addressing the unique needs of PSAI. These discussions will lead to the identification of relevant research priorities for PSAI in Canada. There will be no limit on the number of uncertainties participants can suggest. Each session will last for at least 60 minutes, with the possibility of scheduling a follow-up session to ensure saturation is reached. All working group sessions will be recorded (with informed consent) to facilitate content analysis. Participants will also be asked to complete a short demographic questionnaire to describe the characteristics of the sample. Informed consent will be obtained from all participants. Each patient partner will receive CAD $50 as compensation for their time and contribution to the project.

## Step 3. Summarizing and verifying research uncertainties

All priorities obtained through the working group sessions will be reviewed and sorted by members of the research team, and subsequently discussed with the steering committee. The review process will involve removing duplicate or overlapping priorities, excluding those that are unrelated, combining similar items, and refining complex suggestions into clearly defined research questions using frameworks such as PICO, where appropriate. The sorting process will include coding and grouping priorities into categories under main themes (e.g., culturally and linguistically sensitive maternity care, mental well-being of PSAI, high-risk pregnancy conditions among PSAI). Each priority will be reworded for clarity and ease of understanding. While unrelated priorities will be excluded from this study, they will be documented for future research and shared with the broader research community through a letter to the editor or a short communication in a journal focused on pregnancy care for immigrants. We anticipate identifying approximately 25–30 unique priorities by the end of this stage. The overarching priorities will then be compared against findings from our own scoping review on perinatal care for PSAI in Canada, as well as existing systematic reviews and clinical practice guidelines. This step will help determine whether each priority has been fully or partially addressed by prior research. Priorities that have been adequately addressed in the last five years will be removed, while unanswered or partially answered questions will move forward to the next stage of the process.

## Step 4. Completing an interim priority setting phase

An interim prioritization process will be conducted to consolidate the "long list" of research questions/priorities into a shorter list for discussion at the final priority-setting workshop. This will take place through an interim priority-setting meeting involving the steering committee and other stakeholders who have participated in the project thus far. During this phase, participating patient partners, clinicians/researchers, and representatives from community and professional organizations will be asked to select and rank their top 10 priorities from the consolidated long list. Each selected priority will be assigned a reverse score based on its rank (e.g., 10 points for highest-ranked, down to 1 point for lowest-ranked). The scores from all participants will then be aggregated, and priorities will be ordered based on their total scores. The top 15–18 priorities, based on these aggregate scores, will advance to the final workshop for further discussion and final consensus building.

## Step 5. Holding a final priority setting workshop

A final priority-setting workshop will be held to systematically reach a consensus on the top 10 research priorities for PSAI in Canada. All participants involved in earlier steps will be invited to a virtual workshop to enhance accessibility and ensure broad participation from across Canada. During the workshop, participants will discuss, refine, and rank the top ten research priorities identified in the previous step. Participants will be informed of the workshop's purpose, potential benefits and risks, their role, and the roles of other participants, including the facilitator and members of the steering committee. Approximately 35–50 participants are expected to attend, including patient partners, researchers/clinicians, and representatives from community and professional organizations. Recruitment will occur through partner organizations, the steering committee's networks, contact information provided during earlier steps, and an open call for participation. No individuals wishing to participate are expected to be excluded. However, participants will be screened to ensure balanced representation across roles and experience, as well as diversity in sex and gender, race and ethnicity, age, and geographic location. The workshop will be held via Zoom and will last approximately two hours. Emotional support will be available through a trained research team member with experience in trauma-informed care and culturally responsive counselling, particularly with immigrant and racialized populations, and can be accessed via telephone or a private Zoom call by any participant experiencing distress. The session will be facilitated by Principal Investigator Dr. Anam Shahil-Feroz, who has prior training and experience in conducting prioritization exercises. A modified nominal group technique will be used to guide consensus-building. Participants will first review the shortlist of research priorities generated in Step 4, then engage in small and large group discussions to clarify and reflect on the meaning and relevance of each priority. Following these discussions, participants will individually rank their top priorities. Rankings will be compiled and shared with the group for final reflection, with the aim of collaboratively finalizing the top ten research priorities to improve the health and well-being of PSAI in Canada.

## Data management and analysis

All data from the working group sessions, as well as the interim and final priority-setting phases, will be recorded using Zoom. These recordings will be automatically transcribed in English. Each transcript will be carefully reviewed for accuracy prior to data analysis. An inductive content analysis approach will be used to analyze the qualitative data collected during the working group sessions. Transcripts will first be read in full to gain an overall understanding of the data. Meaningful segments will then be identified and coded, allowing patterns and recurring ideas to emerge organically from the data without imposing preconceived categories. These codes will be grouped into preliminary categories, which will be iteratively refined through constant comparison to ensure internal consistency and clarity. Two members of the research team, both with formal training in qualitative methods, will independently code an initial subset of transcripts to enhance analytical rigor. Discrepancies will be discussed and resolved through

consensus, and a coding framework will be developed and applied to the remaining data. NVivo software will be utilized to facilitate efficient coding, organization, and retrieval of data. Throughout the analysis, reflexive memos will be used to document analytical decisions and enhance transparency. Representative quotes will be selected to illustrate key themes and ensure that participants' voices are authentically represented. The preliminary analysis will be reviewed collaboratively with the broader research team to enhance credibility through multiple perspectives. Additionally, findings will be shared with the project's steering committee for feedback and interpretive validation, strengthening the trustworthiness and relevance of the results. All qualitative data will be anonymized and securely stored. Participant identities will be protected through the use of pseudonyms and removal of identifying details to ensure confidentiality.

## Ethics

Ethical approval for this project has been obtained from the Western University Health Sciences Research Ethics Board (HSREB) (Protocol #126760). All eligible participants will receive detailed information about the study, including its objectives, potential risks and benefits, procedures, the nature of their involvement, and their rights such as the ability to withdraw at any time without consequence. All participants will provide written informed consent. This study involves a PSAI which is a vulnerable group. To ensure ethical inclusion and protection, all consent materials will be written in plain English, and the consent form will be explained verbally in plain English language to support comprehension. Study data will be stored securely on encrypted, password-protected servers at Western University. Data will be retained for three years or until the results are published, whichever comes first. After this period, data will be permanently deleted following institutional data destruction protocols. Only authorized members of the research team will have access to study data. Demographic and key characteristics of all stakeholder groups will be tracked to ensure diverse and representative participation, with particular attention to including voices from vulnerable populations (PSAI) within patient groups. Patient partners will receive CAD $50 as compensation for their time and contribution to the project.

## Dissemination

Following the identification of the top 10 research priorities, several strategies will be employed to raise awareness among researchers, clinicians, PSAI, family caregivers, community and professional organizations, and funders, with the goal of generating new funding proposals that address these priorities. Planned strategies include presentations at national conferences such as the Canadian Conference on Global Health; open-access publication of the top 10 priorities in a peer-reviewed journal for the scientific community; and the creation of a lay summary for community organizations. Media and multimedia tools, such as videos and infographics, will be developed and disseminated to reach PSAI and their family caregivers. Community and professional organizations will also support dissemination through their networks and social media platforms. In addition, findings will be shared with funders through targeted proposals. This prioritization exercise is expected to foster long-term partnerships with patient partners, researchers, clinicians, and community organizations, ultimately leading to the development of a future CIHR Project Grant proposal. This proposal will focus on culturally and linguistically sensitive interventions to support PSAI in Canada.

## Strengths and limitations

A key strength of this study is the culturally sensitive approach embedded throughout the research process. Recruitment will involve trusted community organizations and healthcare providers to build rapport and encourage meaningful participation among South Asian immigrant women. Additionally, working group sessions will be facilitated by team members

and patient partners who have lived experience and cultural competence relevant to the South Asian immigrant communities. Discussion guides will be developed in collaboration with community representatives to ensure cultural appropriateness and relevance. Participant materials, including consent forms and information sheets, will be written in plain English and verbally explained to enhance understanding and accessibility.

One limitation of this study is the potential for selection bias due to the voluntary nature of participation by PSAIs and the reliance on existing relationships for recruitment, which may lead to the under representation of certain subgroups. To mitigate this, we will recruit participants from three provinces including Ontario, British Columbia, and Alberta, using multiple channels, including community organizations, healthcare settings, and online platforms. We will also employ purposive sampling to ensure diversity across key characteristics such as immigration status, education level, employment status, age, number of pregnancies in Canada (parity), and length of time since arrival. This strategy is intended to capture a broad range of perspectives and reduce the likelihood of a homogeneous sample. A second limitation is that the study focuses on PSAI from four South Asian countries including Bangladesh, India, Pakistan, and Sri Lanka. While this scope reflects current demographic trends and the feasibility of the project, we acknowledge that it excludes other South Asian communities. Future research should aim to include these underrepresented groups to ensure broader inclusivity. Finally, only participants with basic English language proficiency will be included in this study. This may exclude the perspectives of PSAI who are not comfortable communicating in English, potentially limiting the inclusiveness and representativeness of our findings. Future studies should consider incorporating interpretation and translation services to enable participation from a wider range of linguistic backgrounds.

## Conclusion

The identification of research priorities and the establishment of a patient-oriented research agenda will help catalyze long-term, impactful research that addresses the unique needs, challenges, and lived experiences of PSAI in Canada. This work holds particular significance for the broader immigrant community by establishing a model for inclusive, culturally, and linguistically responsive maternity care that aims to reduce disparities and improve health outcomes for immigrant groups.

Applying the JLA approach to research priority-setting in partnership with stakeholders is expected to effectively identify and prioritize the unanswered questions that matter most to PSAI and their care providers. To support the uptake of these priorities, efforts will be made to engage researchers, clinicians, community organizations, funders, and government bodies in targeted initiatives for this population. These efforts will include peer-reviewed publications for academic audiences; presentations and policy briefs summarizing key priorities and actionable recommendations for decision-makers and funding agencies; knowledge exchange forums and dissemination events for community organizations and professional associations; and accessible multimedia resources, such as animated videos and voice messages, for patient partners. Additionally, decision-makers and funders will be invited to participate in stakeholder advisory groups or round table discussions to facilitate ongoing dialogue and collaborative planning.

To our knowledge, the JLA approach has not yet been used to identify research priorities specifically for immigrant populations, highlighting the novelty and potential contribution of this study. Our findings are expected to demonstrate the feasibility of applying this structured, inclusive approach to a minority immigrant group (PSAI), which differs from its typical use in disease-specific contexts. Importantly, this work establishes a scalable and adaptable model for inclusive priority-setting that can be extended to other immigrant and marginalized communities, ensuring their unique needs and perspectives are meaningfully reflected in future health research agendas.

## Supporting information

**S1 File. Semi-structured Guide for Working Group Session with Clinicians and Researchers.**
(PDF)

**S2 File. Semi-structured Guide for Working Group Session with Patient Partners.**
(PDF)

**S3 File. Semi-structured Guide for Working Group Session with Community Organizations.**
(PDF)

## Author contributions

**Conceptualization:** Anam Shahil-Feroz, Salima Meherali, Saraswathi Vedam, Anushka Ataullahjan, Rohan D'Souza.

**Funding acquisition:** Anam Shahil-Feroz, Salima Meherali, Saraswathi Vedam, Anushka Ataullahjan, Rohan D'Souza.

**Investigation:** Anam Shahil-Feroz, Saraswathi Vedam, Anushka Ataullahjan, Rohan D'Souza.

**Methodology:** Anam Shahil-Feroz, Saraswathi Vedam, Anushka Ataullahjan, Rohan D'Souza.

**Project administration:** Anam Shahil-Feroz, Bibi Hajira, Sugandha Chandak, Anushka Ataullahjan, Rohan D'Souza.

**Resources:** Rohan D'Souza.

**Supervision:** Anam Shahil-Feroz, Salima Meherali, Saraswathi Vedam, Rohan D'Souza.

**Writing – original draft:** Anam Shahil-Feroz.

**Writing – review & editing:** Anam Shahil-Feroz.

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
