## [Decision Letter · Decision Letter 0]

18 Jul 2025

Dear Dr. Shahil-Feroz,

Thank you for submitting your manuscript to PLOS ONE. After careful consideration, we feel that it has merit but does not fully meet PLOS ONE’s publication criteria as it currently stands. Therefore, we invite you to submit a revised version of the manuscript that addresses the points raised during the review process.

We look forward to receiving your revised manuscript.

Kind regards,

Ahmad H. Al-Nawafleh, Ph.D, MPA, CI, RN

Academic Editor

PLOS ONE

Journal Requirements:

2. Please provide additional information regarding the considerations  made for the pregnant South Asian immigrants included in this study. For instance, please discuss whether participants were able to opt out of the study and whether individuals who did not participate receive the same treatment offered to participants.

3. Thank you for stating the following financial disclosure: [This research is funded by the Canadian Institutes of Health Research Planning and Dissemination Grant (Application No. 197315).]. 

Additional Editor Comments:

Dear authors

Thank you for this interesting topic. I am attaching for you the comments from 3 reviewers. Please consider these comments and revise your manuscript. Probably you need to revise the title too.

Looking forward to receiving your resubmition.

Reviewers' comments:

Reviewer's Responses to Questions

**Comments to the Author**

1. Does the manuscript provide a valid rationale for the proposed study, with clearly identified and justified research questions?

Reviewer #1: Yes

Reviewer #2: Yes

Reviewer #3: Yes

2. Is the protocol technically sound and planned in a manner that will lead to a meaningful outcome and allow testing the stated hypotheses?

Reviewer #1: Yes

Reviewer #2: Yes

Reviewer #3: Yes

3. Is the methodology feasible and described in sufficient detail to allow the work to be replicable?

Reviewer #1: Yes

Reviewer #2: Yes

Reviewer #3: Yes

4. Have the authors described where all data underlying the findings will be made available when the study is complete?

Reviewer #1: No

Reviewer #2: Yes

Reviewer #3: Yes

5. Is the manuscript presented in an intelligible fashion and written in standard English?

Reviewer #1: Yes

Reviewer #2: Yes

Reviewer #3: Yes

You may also provide optional suggestions and comments to authors that they might find helpful in planning their study.

Reviewer #1: Dear author ,

I appreciate the chance to contribute to the evaluation of this work. Below you will find my detailed assessment, which I hope will be helpful in your research process

Reviewer #2: Thank you for the opportunity to review this valuable work. I would like to offer a few comments that may help enhance the clarity and quality of the study

Introduction

1.The Introduction effectively outlines the broader issue of inadequate prenatal care for immigrants and PSAIs. However, the central research gap—the lack of prioritized research on PSAIs’ unique needs—could be stated more clearly and earlier. Consider introducing the research gap in the first 2–3 paragraphs to orient the reader.

2.The term "Pregnant South Asian Immigrants (PSAI)" is introduced midway through the Introduction. Consider defining this term earlier and using it consistently throughout the section to improve readability.

3.Ensure clarity when referring to "immigrants," "newcomers," and "racialized groups" to avoid conflating different populations. Precision in language will aid in scholarly accuracy.

4.The use of references is comprehensive and relevant. However, some references are clustered in long numeric chains (e.g., 4–7; 8–13), which can obscure individual contributions. If feasible, highlight one or two key studies to provide depth rather than breadth.

5.The manuscript mentions "clinical observations from clinicians" (line 20) without referencing specific studies or institutional data. Consider supporting this statement with documented clinical insights or remove it to maintain academic rigor.

6.The brief review of nine studies (lines 25–37) is informative but would benefit from clearer synthesis. Presenting the findings thematically (e.g., structural barriers, cultural mismatches, health literacy) could help the reader better grasp the major barriers faced by PSAIs.

7.While demographic data about South Asians in Canada is helpful, the rationale for focusing specifically on this group (as opposed to other immigrant populations) could be strengthened. Are there known differences in outcomes or health system interactions compared to other immigrant groups?

Material & Methods:

1.While the JLA framework is briefly justified (as “inclusive” and “partnership-driven”), further elaboration on why it is specifically suitable for this cultural and immigrant context would enhance the rationale. Has the JLA model been previously applied with immigrant or minority populations in Canada or internationally?

2.The timeline is clearly stated, but the narrative format could benefit from a table or figure to illustrate phases, activities, and responsible parties. This would help readers visualize the project structure and timelines.

3.The section mentions “South Asian immigrant women with pregnancy experience in Canada within the last 5 years” but does not specify further eligibility criteria (e.g., age range, language proficiency, immigration status). Clarifying these parameters would improve replicability and transparency.

4.It's unclear whether inclusion is limited to cisgender women or if the study is inclusive of gender-diverse individuals. This should be explicitly stated.

5.Although the authors rightly state that formal sample size calculations are not needed for JLA projects, it would be helpful to discuss how diversity will be ensured across subgroups (e.g., different South Asian communities, provinces, socioeconomic status).

6.Since many participants are being recruited through existing relationships, a brief discussion of potential selection bias and how it will be mitigated would be appropriate.

7.The use of semi-structured guides and Zoom-based sessions is appropriate and well-justified given geographic dispersion. However, it would be useful to include an example topic or question from the guide in an appendix or supplementary material to increase transparency.

8.You mention “content analysis” but do not explain the approach (e.g., inductive or deductive coding, use of software like NVivo, coder training, inter-rater reliability checks). This should be described more fully, even if analysis is limited to sorting and categorizing uncertainties.

9.Informed consent and honoraria for participants are well addressed. However, there is no mention of ethics approval (e.g., institutional review board/ethics committee) or whether it is in progress. This should be explicitly stated.

10.The plan for emotional support during the final workshop is commendable. Consider briefly describing what qualifications or training the support person will have.

11.he ranking and scoring methodology during the interim phase could benefit from clarification. Will participants rank all priorities or select a subset? Will average scores or a specific consensus threshold determine advancement?

12.For the final workshop, more information on how consensus will be operationalized (e.g., nominal group technique, majority vote) would strengthen methodological clarity.

13.he text occasionally shifts between future and past tense (e.g., "a committee was formed" vs. "participants will be invited"). Consistency in verb tense would improve readability.

14.The phrase “refining complex suggestions into clearly defined research questions” could be clearer if you mention whether these will be phrased using frameworks like PICO or similar.

15.Consider citing one or two recent JLA-based studies in perinatal or immigrant health to show alignment with best practices.

Reviewer #3: Dear Authors,

This manuscript presents a well-structured protocol for identifying research priorities for pregnant South Asian immigrants (PSAIs) in Canada using the James Lind Alliance (JLA) Priority Setting Partnership (PSP) methodology. The topic is timely, relevant, and important given the increasing diversity in Canada and the documented disparities in maternal health outcomes among immigrant populations. I have some comments as follows:

Introduction

The literature review could be more comprehensive.

Expand on the existing literature regarding maternal health disparities among PSAIs. Consider citing more recent Canadian studies or reports.

Include more recent literature on immigrant maternal health outcomes and their integration into healthcare systems.

Methods

While the lack of a formal sample size calculation aligns with JLA methods, providing a brief justification or referencing similar studies could enhance methodological clarity.

Clarify how funders and policymakers will be engaged post-dissemination to ensure the uptake of research priorities into practice or policy.

Conclusion

Emphasize the potential policy implications and the scalability of the JLA approach to other immigrant communities.

**Do you want your identity to be public for this peer review?** For information about this choice, including consent withdrawal, please see our Privacy Policy

Reviewer #1: No

Reviewer #2: No

Reviewer #3: **Yes: ** Forough Mortazavi

---

## [Author Response · Author response to Decision Letter 1]

29 Jul 2025

COVER LETTER

Manuscript ID: PONE-D-25-29272

Title: Identifying Research Priorities for Pregnant South Asian Immigrants in Canada: A James Lind Alliance Approach

Dear Dr. Ahmad H. Al-Nawafleh,

Thank you for the opportunity to revise and resubmit our manuscript to PLOS ONE. We are grateful for the thoughtful comments and constructive feedback provided by the editor and reviewers. We have carefully considered all suggestions and revised the manuscript accordingly.

Please find enclosed a detailed point-by-point response to the editor and reviewers’ comments in the table below.

As requested, we have included the following statement regarding the role of the funder in our study: “The funders had no role in study design, data collection and analysis, decision to publish, or preparation of the manuscript.”

We appreciate your consideration of our manuscript and look forward to the possibility of its publication in PLOS ONE.

Sincerely,

Anam Shahil-Feroz (she/her)

Assistant Professor

Arthur Labatt Family School of Nursing, Faculty of Health Sciences

FIMS & Nursing Building, Room 3338

Western University, London, ON, Canada N6A 5B9

https://www.uwo.ca/fhs/nursing/about/faculty/research_supervisors/anam_s.html

---

## [Editor Report · Decision Letter 1]

5 Aug 2025

Identifying Research Priorities for Pregnant South Asian Immigrants in Canada: A James Lind Alliance Approach

PONE-D-25-29272R1

Dear Dr. Shahil-Feroz,

We’re pleased to inform you that your manuscript has been judged scientifically suitable for publication and will be formally accepted for publication once it meets all outstanding technical requirements.

Kind regards,

Ahmad H. Al-Nawafleh, Ph.D, MPA, CI, RN

Academic Editor

PLOS ONE

Additional Editor Comments (optional):

Dear authors,

Thank you for your response to the comments and the careful consideration.

Looking forward to read the results of your study after it's implementation.

The topic is important and I hope you will apply it on other groups of migrant women from other regions of the world.

All the best
---

## [Editor Report · Acceptance letter]

PONE-D-25-29272R1

PLOS ONE

Dear Dr. Shahil-Feroz,

I'm pleased to inform you that your manuscript has been deemed suitable for publication in PLOS ONE. Congratulations! Your manuscript is now being handed over to our production team.

Kind regards,

on behalf of

Dr. Ahmad H. Al-Nawafleh

Academic Editor

PLOS ONE